# Social Robot Navigation Tasks: Combining Machine Learning Techniques and Social Force Model

**DOI:** 10.3390/s21217087

**Published:** 2021-10-26

**Authors:** Óscar Gil, Anaís Garrell, Alberto Sanfeliu

**Affiliations:** Intitut de Robòtica i Informàtica Industrial (CSIC-UPC), Llorens i Artigas 4-6, 08028 Barcelona, Spain; oscar.gil.viyuela@upc.edu (Ó.G.); anais.garrell@upc.edu (A.G.)

**Keywords:** social robot navigation, Social Force Model, Reinforcement Learning

## Abstract

Social robot navigation in public spaces, buildings or private houses is a difficult problem that is not well solved due to environmental constraints (buildings, static objects etc.), pedestrians and other mobile vehicles. Moreover, robots have to move in a human-aware manner—that is, robots have to navigate in such a way that people feel safe and comfortable. In this work, we present two navigation tasks, social robot navigation and robot accompaniment, which combine machine learning techniques with the Social Force Model (SFM) allowing human-aware social navigation. The robots in both approaches use data from different sensors to capture the environment knowledge as well as information from pedestrian motion. The two navigation tasks make use of the SFM, which is a general framework in which human motion behaviors can be expressed through a set of functions depending on the pedestrians’ relative and absolute positions and velocities. Additionally, in both social navigation tasks, the robot’s motion behavior is learned using machine learning techniques: in the first case using supervised deep learning techniques and, in the second case, using Reinforcement Learning (RL). The machine learning techniques are combined with the SFM to create navigation models that behave in a social manner when the robot is navigating in an environment with pedestrians or accompanying a person. The validation of the systems was performed with a large set of simulations and real-life experiments with a new humanoid robot denominated IVO and with an aerial robot. The experiments show that the combination of SFM and machine learning can solve human-aware robot navigation in complex dynamic environments.

## 1. Introduction

The introduction of robotics in our daily activities in the near future will require the navigation of humans and robots in different environments, including public spaces, buildings and private houses. The applications are huge, for example in assistive robotics tasks [1], collaborative searching [2], side-by-side navigation [3], guiding people [4] or other social navigation tasks. In any of these applications, robots have to behave in a social manner, being social-aware (that is, the robot has to plan the trajectory helping the accompaniment or guiding a person while not disturbing other pedestrian trajectories if possible) and avoiding colliding with obstacles or pedestrians.

In this work, we present a combination of SFM and machine learning techniques that, using the robot’s perception systems, make the navigation of the robots easier in environments where there are static obstacles, such as walls or urban furniture and moving objects, like pedestrians or bicycles. The SFM is a general framework in which the human motion behaviors can be expressed through a function depending in the pedestrians relative and absolute positions and velocities.

In machine learning techniques, we present two different approaches: a supervised learning approach based on a neural network that is used for accompanying a person with an UAV (Unmanned Aerial Vehicle) and an RL technique that is used for robot navigation in areas where there are pedestrians moving around.

The SFM and the machine learning techniques are combined in a different ways in each of the applications. In the first case, the forces of the SFM are the inputs of a neural net, and the neural network (NN) learns the combination of repulsive forces produced by the environment and the pedestrian motions. In the second case, the SFM is used to avoid the static and motion obstacles, and the learned robot motion force is combined with the repulsive forces produced by the environment and pedestrian motions. In both cases, the resulting robot motion force is human-aware, because it modifies the robot behavior based on the pedestrian motion trajectory.

These two techniques were tested in simulation and real-life experiments. In Social Robot navigation experiments, simulation analysis has been done in complex urban scenarios and real-life experiments were performed by the IVO robot (see Figure 1). In the IVO’s robot, we use two types of sensors, a 3D LiDAR to detect obstacles, pedestrian motions and to allow the self-localization of the robot as well as an RGBD RealSense camera to detect holes and ramps. Moreover, we use the odometry’s robot sensors that are combined with the LiDAR localization sensor to obtain robustness in the robot localization.

In the UAV human accompaniment, the simulation analysis was performed in a simulated environment that included trees and walls, and the real-life experiments were performed between an UAV and a person, where the UAV accompanied the person in an environment with static obstacles. We used external Optitrack sensors to detect the human motion and the robot motion.

In this work, we describe in Section 2, a summary of the state of the art in social robot navigation, social robot accompaniment and deep RL in robot navigation. In Section 3, we introduce the problem definition. In Section 4, we describe the SFM and Automated Reinforcement Learning (AutoRL). In Section 5, we introduce the combination of RL with SFM, and, in Section 6, we describe the combination of supervised learning with SFM. Section 7 and Section 8 describe the simulation analysis and real-life experiments of the social robot navigation and UAV human accompaniment tasks. Finally, Section 9 relates our conclusions.

## 2. State of the Art

The overall goal of this research is to develop robots that work cooperatively with humans while developing navigation tasks. As such, this article draws on work from diverse fields, including social robot navigation and the social robot accompaniment of people. This section introduces the data yielded by some of the most relevant research on this subject.

### 2.1. Social Robot Navigation

Human motion social interaction has been studied previously in several works [5,6,7]. Deepening on the SFM introduced by Helbing & Molnar in [8], human motion is modeled as a reaction to a set of attractive and repulsive forces generated by other people or the elements present on the environment, where people move safely and more comfortable along the minimum potential paths of the forces field.

In social navigation, a commonly used metric is the proxemic rules between humans and robots; moreover, Walter et al. [9] analyzed how the personality affects pedestrian spatial zones. Alternatively, another approach studied the person–robot proxemic distances in different interactions: verbal, physical and no interaction [10]. Furthermore, in [11,12], researchers found that people prefer robots to approach them by one of their sides; for instance, right-handed people prefer to be approached from the right.

Most social navigation models have either strict situational applicability or a pronounced reactive nature, as models defined based on the SFM stated by Helbing & Molnar [8]. Due to this, some models included predictive models on their structure [13,14] or assume the knowledge of the location of the final goal of the pedestrian or robot trajectory [15], which might demand a high computation cost and confer them a heuristic nature. The model proposed attempts to learn human behavior from a static world distribution, not relying on any knowledge inference. It aims to implement implicit prediction through the reading of the environmental characteristics.

Common walking habits of humans were investigated, as individuals, pairs or groups in [16,17,18,19,20]. Garrell et al. [13] focused on side-by-side pair walking, whose line of work is followed along this article. Other interesting approaches of social navigation are applied to autonomous wheelchairs [20], some of them focused on side-by-side movement [21,22].

This work goes one step forward, developing a new method based on Multi-Layer Perceptron to improve human–robot side-by-side social navigation. To tackle the introduced problem, we propose a new nonlinear-based approach that uses the Extended Social Force Model (ESFM) [23] as a means to extract meaningful information from the environment.

The field of Human Motion Prediction is similar to that of Social Navigation with the difference that only attempts to predict the future trajectories of humans in a scene, not navigate around them. Recent works in this field have focused on predicting future trajectories using Recurrent Neural Networks (RNNs) to encode given sequences of positions [24] and poses [25] and map them to predicted trajectories. These publications have improved works in the field; however, these models do not consider the Multimodality aspect in human trajectories, which is why works, such as [26,27] use Generative Adversarial Networks (GAN) in order to solve these problems. This approach to train the encoder–decoder RNN models, allows us to focus on our objective of obtaining correct socially acceptable trajectories instead of attempting to approximate the ones already in the database.

A common trait among all these methods is that, in order to obtain information and learn to account for social norms in their predictions, all these works keep a model of every person in a scene and process them together inside a self-designed pooling layer. This allows obtaining compact representations of the environment to feed into the model. However, these procedures are usually computationally expensive and difficult to scale up to a huge number of humans in a scene. Additionally, these models only take into consideration the humans present in the scene; only [27] attempted to account for static objects in the scene by feeding information on the top view of the scene to the model. However, this top view is not always available.

Seeing the limitations of these procedures, we propose the use of the SFM [8] as an alternative to obtain simple and compact representations of the environment. Given the position of static objects and humans in a region centered around the robot, the SFM allows us to describe in a simple and scalable way the environment without the need for external information.

### 2.2. Deep Reinforcement Learning in Robot Navigation

In recent years, deep RL algorithms [28,29] have demonstrated human-level control in some tasks, such as video games [30,31], where this is almost impossible to obtain using handcrafted features for the states or considering fully observable environments. For this reason, deep RL approaches are very common for visual navigation [32,33].

Other deep RL models are based on different environment features. In [34,35], a fully-observable environment with a discrete set of actions is considered where the robot receives all the agent positions and velocities. In [35], a Long Short-Term Memory (LSTM) cell is fed with an arbitrary number of moving agent states, and a training is performed considering all the agent experiences in a scene to increase the joint expected reward. In [34], the moving agents are simulated using the Optimal Reciprocal Collision Avoidance algorithm (ORCA) [36], and Imitation Learning is used to improve the performance. Although, as is described in [37], the performance is only high when the robot is close to the goal with few moving obstacles around.

Some works consider prediction information to improve the algorithm, for example in [38], where people are tracked, and the next frame for a person is predicted. This prediction is used as an additional observation in the Proximal Policy Optimization algorithm (PPO). This prediction reduces the freezing robot problem in dense crowds.

## 3. Problem Definition

As has been mentioned previously, in this work, we investigate two different social navigation tasks, social robot navigation and robot accompaniment, and both combine machine learning techniques with the Social Force Model (SFM).

In this section, we proceed to describe the problem definition of both social tasks.

### 3.1. Reinforcement Learning with Social Force Model

Given a dynamic environment, which consists of static objects and pedestrians, the objective is to navigate being human-aware. The robot actions, linear and angular velocities, are computed by a combination of robot velocities learned by a reinforcement learning model (AutoRL [39]) and robot velocities computed using a SFM. The SFM is used to take into account the moving pedestrians, which allows human-aware navigation. In this model, we use robot Lidar measurements to acquire the environment and pedestrian observations.

We consider the robot navigation task as a Partially Observable Markov Decision Process (POMDP) defined by the tuple (O,A,D,R,γ) where the action space, *A*, and the observation space, *O*, is considered continuous. *D* are the dynamics, provided by the simulation environment or the real world, *R* is the reward described in Equation (Equation 13), and γ∈(0,1) is the discount factor.

The robot model is a circle of radius 0.3 m, and the actions are the typical velocity commands: linear and angular velocities, $a=(a_{l},a_{\phi })$. The observations, $o={\left(o_{l},o_{g}\right)}^{\theta_{n}}\in O$, are divided in two parts: $o_{l}$ are 64 1-D LiDAR distance lectures taken during the last $\theta_{n}$ simulation steps and $o_{g}$ are the goal polar coordinates during the last $\theta_{n}$ steps. Each simulation step is about 0.2 s, and the LiDAR field of view is $220^{{}^{\circ}}$. The distance range for the LiDAR lectures goes from 0 to 5 m. The action space for the linear velocity is $v_{l}\in \left[-0.2,1\right]$ m/s and $v_{\phi }\in \left[-1,1\right]$ rad/s for the angular velocity.

### 3.2. Supervised Learning with Social Force Model

Given a dynamic environment, composed of both static and mobile/aerial social entities, the objective is to accompany a determined actor in the environment in a socially acceptable manner, both on task and social navigation perspectives. As introduced before, this problem is approached from an egocentric perspective, entailing a strict restriction on world perception. It is important to remark on mobile social actors in the environment may change their behavior influenced by the robot movement.

As a result, all robot interactions with the different elements of the environment are integrated into four different forces. These forces represent the features that the learning model uses as input to predict the acceleration commands that must be applied to the robot, which are defined, in detail, in the following subsection. The presented research can be applied to any type of robots, nevertheless, for the current article, we will focus on aerial robots.

## 4. Background

In the present section, we proceed to describe the background methods used in this article, which are the SFM and the AutoRL approach for Robot Navigation.

### 4.1. Social Force Model

In the late 1950s, pedestrian behaviors were modeled by the first time. At the beginning, these models were concentrated on the dynamics of macroscopic theories, and pedestrian dynamics were studied as fluids [40]. As the development advanced, scientist focused more on microscopic description, here, the motion of each pedestrian is described independently [41].

In contrast, the SFM [8] reproduces pedestrian dynamics using interaction forces. Thus, the human motion behavior can be described as a function depending on the pedestrians’ positions and velocities. Furthermore, authors presented in [19] combined the time of collision with the SFM. Nevertheless, these works do not take into account the interactions between robots and people. For the representation of these interactions, we were influenced by the works of [8,19].

Formally, this approach considers each pedestrian $p_{i}$ in the environment with mass $m_{pi}$ as a particle following the laws of Newtonian mechanics:(1)$${\left[\begin{array}{c} x\\ y\\ v_{x}\\ v_{y} \end{array}\right]}_{t+1}=\left[\begin{array}{cccc} 1 & 0 & \Delta t & 0\\ 0 & 1 & 0 & \Delta t\\ 0 & 0 & 1 & 0\\ 0 & 0 & 0 & 1 \end{array}\right]{\left[\begin{array}{c} x\\ y\\ v_{x}\\ v_{y} \end{array}\right]}_{t}+\left[\begin{array}{cc} \frac{\Delta t^{2}}{2} & 0\\ 0 & \frac{\Delta t^{2}}{2}\\ \Delta t & 0\\ 0 & \Delta t \end{array}\right]\left[\begin{array}{c} a_{x}\\ a_{y} \end{array}\right]$$
where (x,y) is person’s position, $\left(v_{x},v_{y}\right)$ is his/her velocity, and $\left(a_{x},y_{x}\right)$ is the acceleration.

Assuming that pedestrian attempts to adapt his or her velocity within a *relaxation time* $k_{i}^{-1}$, $f_{i}^{goal}$ is given by:(2)$$f_{i}^{goal}=k_{i}\left(v_{i}^{0}-v_{i}\right)$$

In order to reach a particular desired velocity and direction a relaxation time is required. Moreover, the force $F_{i}^{int}$ represents the repulsive effects caused by obstacles, pedestrians and robots in the environment. This force is conformed as an addition of forces introduced by people $p_{j}$, by static obstacles in the environment *o* or by a particular robot *R*.
(3)$$F_{i}^{int}=\sum_{p_{j}\in P}f_{ij}+\sum_{o\in O}f_{io}+\;f_{iR}$$
where, P is the set of pedestrians in the environment, and O is the set of static obstacles.

Usually, people maintain certain distances from other humans in the environment. Pedestrians feel uncomfortable the closer they are to an unknown person.
(4)$$f_{ij}=Ae^{(d-d_{ij})/B}\frac{r_{i,j}\left(t\right)}{d_{i,j}\left(t\right)}$$
where the set of parameters {A,B,d} denotes the strength, and the range on interaction force, respectively. $r_{i,j}=r_{p_{i}}-r_{p_{j}}$ is the sum of radius of the two pedestrians involved in an interaction. See [3] for further details.

In addition, an obstacle *o* creates a virtual repulsive effect, as people feel less comfortable the closer an obstacle is navigated, and this can be expressed as:(5)$$f_{i,o}=\nabla_{r_{i}o}U_{io}^{0}e^{\left|\right|r_{io}\left|\right|/C}$$
where $r_{io}=r_{p_{i}}-r_{io}$ represents the location of the point of the obstacle *o* that is closest to pedestrian $p_{i}$ [42] for a more detailed explanation.

Lastly, people maintain a security distance from robots. Thus, a robot *R* creates a repulsive effect if the distance to the human is lower than a particular threshold. This effect is expressed as:(6)$$f_{iR}=A_{iR}e^{\left(d_{r}-d_{iR}\right)/B_{iR}}\frac{r_{iR}\left(t\right)}{d_{iR}\left(t\right)}\left(\lambda_{iR}\left(1-\lambda_{iR}\right)\frac{1+cos(\phi_{iR})}{2}\right)$$

The described theory is used to build a social robot navigation tasks framework; here, the robot moves naturally in human environments following the SFM, and thus we achieve a higher acceptance from pedestrians.

### 4.2. Social Force Model with Collision Prediction

The SFM with Collision Prediction, described in [19], is a model that takes into account the velocities of all the agents in an environment and makes a prediction of their future collisions. These future possible collisions and these agent’s velocities are used to compute the repulsive forces applied in all the agents.

To estimate the future collisions, the agents are propagated in time with a constant velocity model. Then, for each pair of pedestrians, *p*, *q*, the time in which the difference between agent positions, $d_{q,p}$ becomes minimal is computed using this formula:(7)$$t_{q,p}=\left\{\begin{array}{cc} -\frac{d_{q,p}^{T}\cdot v_{q,p}}{∥v_{q,p}{∥}^{2}} & |\theta_{q,p}|<\pi /4\\ \infty & |\theta_{q,p}|>\pi /4 \end{array}\right.$$
where $v_{q,p}$ is the relative velocity between *p* and *q*, and $\theta_{q,p}$ is the angle between $v_{q,p}$ and $d_{q,p}$.

For each pedestrian *p*, there is a set of times in which a possible collision can occur, $\left\{t_{q,p}\right\}$. To estimate a possible collision, the most important time of the set to be considered is the minimal one, $t_{p}$:(8)$$t_{p}=min_{q}\left\{t_{q,p}\right\}$$

Using this time estimation, the repulsive force expression applied in agent *p* due to the agent *q* is:(9)$$f_{q,p}^{\mathrm{int}}\left(\left\{v_{q,p}\right\},\left\{d_{q,p}\right\},v_{p}\right)=A_{q}\frac{v_{p}}{t_{p}}e^{-d_{q,p}/B_{q}}\frac{d_{q,p}^{\prime }\left(t_{p}\right)}{d_{q,p}^{\prime }\left(t_{p}\right)}$$
where $\left\{v_{q,p}\right\}$ is the set of relative velocities between *p* and other agents *q*. $\left\{d_{q,p}\right\}$ is the set of vectors with all relative distances between *p* and other agents, *p*. $v_{p}$ is the velocity module for *p*. $d_{q,p}^{\prime }\left(t_{p}\right)$ is the relative position of *p* regarding *q*, in *t* = $t_{p}$. $A_{q}$ and $B_{q}$ are parameters that can be adjusted and $d_{q,p}$ is the Euclidean distance between agents.

This repulsive force can be applied to static obstacles considering zero velocity for each obstacle *q*. In this model, there is an attractive force, given by Equation (Equation 2), to encourage the agent to achieve a goal.

The resultant force for each agent is the sum of the repulsive forces from other agents, from static obstacles and the attractive force to the goal:(10)$$F_{p}=f_{p}^{\mathrm{goal}}+\sum_{q\in P}f_{q,p}^{\mathrm{int}}+\sum_{o\in O}f_{o,p}^{\mathrm{int}}$$
where *P* is the set of agents and *O* is the set of static obstacles. This model is very useful to simulate the pedestrian behavior near the obstacles and to avoid collisions, maintaining the direction toward the goal. However, this model is not so good when faces large static obstacles.

### 4.3. AutoRL for Robot Navigation

AutoRL [39] is an algorithm that searches the optimal hyperparameters for a parametrized reward and for the neural network in deep RL. For a feed-forward fully connected network, the parameters are the number of hidden layers and the number of neurons for each hidden layer. In [39], these optimal hyperparameters are searched and used through the DDPG algorithm for robot navigation in two navigation tasks: path following (PF) and point-to-point (P2P).

The PF task consists on following a guidance path obtained through the method described in [43]. The P2P task consists on navigating from an initial position to a goal. To find the best hyperparameters, AutoRL maximizes an objective function *G* that is evaluated for each training with different sets of hyperparameters. This function depends on the task:(11)$$G_{P2P}\left(s\right)=I(∥s-s_{g}∥<d_{P2P})$$
(12)$$G_{PF}\left(s\right)=\frac{\sum_{\omega \in P}I(∥s-\omega ∥<d_{wr})}{∥P∥}$$
where *s* is the robot position and I is the indicator function. For P2P, $s_{g}$ is the goal position and $d_{P2P}$ is the goal radius. For PF, ω are the way point positions of the guidance path P and $d_{wr}$ is the way point radius. The objective in P2P is to maximize the probability of reaching the goal during an episode. The objective in PF is to reach as many way points as possible in each episode.

For the P2P task, the parametrized reward proposed in [39] is:(13)$$R_{\theta_{r_{P2P}}}=\theta_{r_{P2P}}^{T}\left[r_{step}\;r_{goalDist}\;r_{col}\;r_{turn}\;r_{clear}\;r_{goal}\right]$$
where $\theta_{r_{P2P}}$ are the reward hyperparameters for the P2P task, $r_{step}$ is the penalty constant at each step in the episode with value 1, $r_{goalDist}$ is the negative Euclidean distance to the goal, $r_{col}$ is 1 in collisions and zero otherwise. $r_{turn}$ is the negative angular speed, $r_{clear}$ is the distance to the closest obstacle and $r_{goal}$ is 1 when the agent reaches the goal and zero in the remaining cases. To know more details of Equations (Equation 11)–(Equation 13), see the work of [39].

In [44], AutoRL was demonstrated to be able to achieve very good results in other tasks compared to hand-tuned and hyperparameter-tuning methods, such Vizier [45]. However, the time required and the computational cost to apply AutoRL can make it difficult to use in cases where there are not enough computational resources.

To extend the navigation distance in large indoor environments, AutoRL has been implemented as a local planner for a PRM [46] or RRT [47]. The results in real environments show a high performance and robustness to noise.

## 5. Combining Reinforcement Learning with Social Force Model

The approach described in this section presents a hybrid policy for social robot navigation, which combines a RL model with the SFM to improve social navigation in environments that contain pedestrians and obstacles. Specifically, the SFM version used for this approach is the one described in [19].

### Hybrid Model Description

This section explains the details of the hybrid model based on AutoRL work for P2P navigation.

First, the model is trained using the DDPG algorithm with the same optimal hyperparameters, obtained with the AutoRL described in [46] for a differential drive robot model. The noise parameters are the same too. Although the task can be seen as a POMDP, it is very common to approximate it as a Markov Decision Process (MDP) to use deep RL algorithms, such as DDPG where the observations are treated as states. This algorithm uses feed-forward networks for both the actor and the critic. The actor network is the policy, with network parameters $\theta_{a}$. The actor takes $\theta_{n}$ observations and gives an action:(14)$$\pi \left(o;\theta_{a}\right)=\left(a_{l},a_{\phi }\right)$$

Secondly, in the evaluation phase, the SFM with collision prediction is applied to each point detected using the LiDAR. Only the repulsive force described in Equation (Equation 9) is considered. The attractive force in (Equation 2) is not considered in the model, because the trained model has learned to efficiently navigate toward the goal when there are not large obstacles close to the robot in the goal’s direction. The objective of using the SFM in this work is to use it only when the robot is close to the obstacles to avoid collisions. The repulsive force is almost zero far from the obstacles.

In simulation, the agent’s velocities used to calculate the repulsive forces are known if the agents are detected with the robot LiDAR. In a real implementation, these velocities can be provided by an efficient tracker, for example, the ones described in [48,49], that can provide agent velocities in real time using the LiDAR observations.

There is a repulsive force for each LiDAR point, $f_{i}$. These set of repulsive forces can be sorted by module from highest to lowest, ${\left\{f_{i}^{\mathrm{sort}}\right\}}_{i=1}^{n}$. To capture the most important interactions, the five largest forces are considered each time-step to calculate a resultant force:(15)$$F_{res}=\sum_{i=1}^{5}f_{i}^{\mathrm{sort}}$$

To avoid very large forces, the resultant force is normalized and multiplied by the biggest force module, $f_{1}^{sort}$, to obtain the final repulsive force:(16)$$F_{rep}=f_{1}^{sort}\frac{F_{res}}{∥F_{res}∥}$$

The module of $F_{rep}$ represents the nearest obstacle influence, and the force direction represents the best direction to avoid the nearest obstacles around the robot.

The force $F_{rep}$ is used to compute the robot’s velocity. Since we consider in the SFM that the robot has mass = 1, then the robot acceleration is the force, and we use this acceleration to compute the changes of velocity ${\left(\Delta v\right)}_{rep}$ in each time-step. This velocity change can be decomposed in 2 components, ${\left(\Delta v\right)}_{rep}=\left(\Delta v_{l},\Delta v_{\phi }\right)$. $\Delta v_{l}$ is the component in the current velocity direction, and $\Delta v_{\phi }$ is oriented in the orthogonal direction. These velocity changes or SFM actions are added to the DDPG actions to calculate the actions, which are provided by the hybrid model:(17)$$a_{l}^{\prime }=a_{l}+\Delta v_{l}$$
(18)$$a_{\phi }^{\prime }=a_{\phi }+\Delta v_{\phi }$$

The weights of the SFM and DDPG’s actions in the Equations (Equation 17) and (Equation 18) are the same, although the SFM only acts when the robot is near to obstacles. The SFM action is almost zero if the distance is more than 1 m, but if the robot is very close to the obstacles, the SFM action is much bigger than the DDPG action.

The complete scheme of the hybrid model is shown in Figure 2.

## 6. Combining Supervised Learning with Social Force Model

The ESFM can be considered as a means to extract a simple representation of the environment useful in social navigation problems. Therefore, given a set of social forces, our goal is to predict—through supervised learning techniques—the velocity vector that will allow the robot to follow the necessary motion to obtain a proactive human-like side-by-side navigation with humans.

For that reason, we present a nonlinear-based approach related to the ESFM. Ahead, we proceed to describe the combination of the supervised learning with the well-known SFM. The proposed system aims to allow human–robot side-by-side social navigation in real environments, concretely, drone navigation, and this approach is based on the extension of the SFM presented in [50].

### 6.1. Feature Definition

SFM has demonstrated that changes in behavior/intentions of humans can be explained in terms of social forces or social fields. The total force applied to the robot comes from the following components: (i) and (ii) the robot-humans and robot-objects interaction forces, which are repulsive, and (iii) the goal attraction force, which is a predicted position that makes the robot stay closer to the human being accompanied. All these forces are incorporated in the non-linear SFM to the set of input features, which also includes instantaneous drone velocity.

#### 6.1.1. Repulsive Features

First, the static object repulsive feature provides information about the relative location of static objects with respect to the robot, and it also offers information about their proximity. It takes the form of a global force that aggregates all the individual repulsive forces:(19)$$F_{o}=\sum_{o=1}^{O}f_{o}$$
(20)$$f_{o}=\frac{P_{o}-P_{R}}{∥P_{o}-P_{R}∥}A_{R,o}e^{\left(\frac{d_{R_{o}}-d_{R,o}}{B_{R,o}}\right)}$$
where *O* is the set of detected objects; $f_{o}$ is defined as a non-linear function of the distance of each detected static objects in the environment; $P_{o}$ and $P_{R}$ are the positions of the object and the drone, respectively; and $d_{R_{o}}$ is the distance between the robot and the object. $A_{R,o}$, $B_{R,o}$ and $d_{R,o}$ are fixed parameters; the study of the definition of the parameters is described in [50].

Secondly, pedestrians can be considered as dynamic obstacles; nevertheless, bystanders have their own personal space, and intrusions into this area by external objects may cause discomfort to the person. Thus, we define a feature to model: the repulsion exerted on the drone from pedestrians. The pedestrian repulsive feature is again a force defined similarly to the static object repulsive force,
(21)$$F_{h}=\sum_{h=1}^{H}f_{h}$$
where *H* is the set of detected pedestrians and $f_{h}$ is defined as
(22)$$f_{h}=\frac{P_{h}-P_{R}}{∥P_{h}-P_{R}∥}A_{R,h}\frac{v_{h}}{t_{h}}e^{\left(\frac{d_{R_{h}}-d_{R,h}}{B_{R,h}}\right)}$$

$P_{h}$ and $P_{R}$ are the positions of the pedestrian and the drone, respectively, and $d_{R_{h}}$ is the distance between the robot and the object. Moreover, the term $v_{h}/t_{h}$ is introduced to model the force in such a way that the pedestrian *h* is able to stop in time $t_{h}$ [19].

#### 6.1.2. Attractive Features

In order to obtain a flying behavior in which the drone is moving in a side-by-side formation with the human, some notion of the path the human and is following is required. This module is responsible for inferring future human’s position.
(23)$$f_{g}=P_{g}-P_{R}$$
where $P_{R}$ is the drone’s position and $P_{g}$ is the estimated human position for one second into the future. This feature is not expressed as a force, but still provides the necessary information to allow the drone to fly next to the main human.

Moreover, a new feature is defined, it provides information about the current position of the human. This feature combined with the goal feature encodes information about the current position of the human and the expected direction of movement. The human feature is defined similarly to the goal feature, as a vector from the drone position $P_{R}$ to the main human position $P_{h}$.
(24)$$f_{c}=P_{c}-P_{R}$$

As can be seen in Figure 3, we plotted all the forces used as features in our supervised learning approaches, moreover, we defined an additional destination to the robot approach. The robot aims to the target person in order to accompany him/her.

### 6.2. Non-Linear Regressor

Social Navigation based on Artificial Potential Fields was presented previously [50], and these works extract forces from the elements in the environment, and combine them linearly obtaining a resulting force. This force is then applied to the robot, which determines its acceleration. The aim of this research is to substitute the linear approach by a non-linear model learned from data, the data is composed by recordings of robot’s trajectories tele-operated by an expert user. We desire to train a dense neural network to imitate the flying controls of a human expert. By learning the model from data, we also avoid the tuning of the presented parameters as in previous approaches.

The non-linear model used to learn the expert flying policy is a fully connected five-layer dense neural network, where the first hidden layer contains 18 neurons, and the second hidden layer is made of 23 neurons. The input layer has 15 neurons, one for each input dimension, and the output layer has three neurons, one for each component of a linear acceleration in a 3D space. This adds up to a total of 797 trainable parameters.

The probability values defined for the network are, from input to output layer, 0.1 and 0.1. Regarding the parameters used during the training phase of the networks, the mean squared error was the chosen loss function and Adam was used as the gradient descend optimizer, and it was trained with batches of size 512 [51].

Validation data, consisting of an episode of each situation, is extracted and separated from the rest. All windows from each episode from the rest of the dataset are then pooled together, scaled, shuffled and split into training and test datasets. Then, the model is trained and tested. Moreover, validation is done in two phases. First, we compare the Root Mean Square Error (RMSE) of the estimated force with the results obtained following previous works [52]. Secondly, we assess the performance of our model by comparing its RMSE with previous works in specific environments using the data from a specific episode of the validation data.

FreeGLUT toolkit, and the joysticks used for user control are PlayStation 3 DualShock 3 Wireless Controllers.

On the knowledge transfer to real-world applicability, it has been chosen to implement over ROS (Robot Operating System). Two steps were taken in this process, first a model testing phase over a simulated *Gazebo* world and, secondly, an execution phase on a *Drone* base. On both, all world data were collected through the processing of two laser inputs -*Hokuyo UTM-30LX Scanning Laser Rangefinder*-.

### 6.3. Quantitative Metric of Performance

To properly evaluate the learned control policies, the Non-linear Aerial SFM defines a new quantitative metric of performance inspired by the metric presented in [23], which is based on the idea of proxemics [6].The new quantitative metric defines three distances, $d_{A}$, $d_{B}$ and $d_{C}$, which define three concentric spherical volumes around a human. Please refer to [23] to check the definition of the used metrics.

## 7. Experiments: Social Robot Navigation

In this section, we explain the simulation and real-life experiments in social robot navigation, using the combination of a Reinforcement Learning technique with the SFM.

For the simulations, we used the maps shown in Figure 4: one of them for training and the rest for evaluation. The training and the evaluation described in this section were performed using the OP-DDPG described in this article with the optimized parameters of the AutoRL work [39].

### 7.1. Metrics

We have used the following three metrics to evaluate method in the maps:SuccessRate (SR): Percent of episodes in which the robot achieves the goal without collision. It is a measure of the robot behavior to move closer to the goal and gives an idea of the overall performance. When the robot reaches the goal, the episode is finalized.CollisionRate (CR): Percent of episodes that ends in a collision. A collision marks the end of an episode.TimeoutRate (TOUT): Percent of episodes in which the episode time limit is reached without a collision or success. This metric gives information in complex environments or very uncommon situations for the robot.

During the execution, other metrics are calculated at each episode as for example, the number of steps and the goal distance to obtain a more local description of each episode.

### 7.2. Evaluation Results

For all the evaluations, we used the same trained OP-DDPG, which was trained with the map of Figure 4a. The hybrid model is introduced in the evaluation phase, where the combination of RL and SFM is applied.

For the evaluation, we used two different types of distances: the Euclidean distance (ED) and the approximate path distance (PD). The main reason to use the approximate path distance is that it provides more precise results of the policy performance when the path to the goal is obtained using PRM [46] or RRT [47] planners. When it is used the ED, the goal is sampled between 5 and 10 m; and when the PD is used, the goal is obtained using a RRT’s path whose length is less than 10 m. The results shown in Table 1, obtained with 5–10 m of Euclidean distance and path distance, show that the hybrid model can increase the success rate in the three evaluation maps without moving obstacles. All the evaluation results were obtained computing 100 episodes as in [39].

The results obtained using Euclidean distance were worse than the obtained ones in [39] as the implementations are different. This means that, when the AutoRL’s optimal hyperparameters are applied in the OP-DDPG model, suboptimal results were obtained. In this work, the results are focused in the advantages of applying the SFM, regardless the AutoRL optimization.

The evaluation with moving agents was also performed using the Euclidean and path distance. There are two cases that represent a low-crowded environment with 20 moving agents (pedestrians) and a high-crowded scene with 100 moving agents.

The results in Table 2 show that the number of moving agents does not greatly affect the hybrid model performance. Only in environments with a very high density of moving obstacles, such as Building 2 with 100 obstacles, is the success rate significantly reduced. The differences in the success rate between the Euclidean and path distance cases increase in the high-crowded cases because the number of encounters with moving agents is augmented.

Another evaluation (refer to Figure 5) was performed increasing the goal path distance in the range from 10 to 15 m. In this case, the hybrid model shows less degradation in the success rate compared with the OP-DDPG model.

Robustness to noise was evaluated in the maps with 20 moving agents. In these cases, such as in the AutoRL work, Gaussian noise is considered, $N(0,\sigma_{lidar})$, for four different standard deviations between 0.1 and 0.8 m. The results obtained show that the OP-DDPG model and the hybrid model are very robust to noise. The success rate decreases in both models less than 5%.

The difference between the OP-DDPG model and the hybrid model lies mostly in the collision rate reduction. On the contrary, the timeout rate can increase, because a percentage of the episodes that lead to collisions in OP-DDPG model become episodes that end in timeouts in the hybrid model. In the hybrid model, the repulsive forces have only influence when the robot is close to the obstacles, due to the exponential decay of those forces. For this reason, the majority of actions are taken by the OP-DDPG policy. This causes the hybrid model to avoid collisions, but the repulsive force of SFM is not enough to escape from complex obstacles if the OP-DDPG’s trained policy has not learned to avoid those situations properly.

The timeout episodes are associate to large scale local minima, for example, a room with a small exit where the robot moves in a loop. The timeouts give information about difficult situations that have not been learned, for example when the robot has to move in the big open spaces in the map building 1. These spaces have not been observed in the training map, leading to worse behavior to reach the goal. That is the reason why timeouts are larger in building 1. Despite timeout increasing cases, the overall performance is better with the hybrid model.

### 7.3. Real-Life Experiments

Real indoor experiments were performed with the IVO robot owned by the Institut de Robòtica i Informàtica Industrial [53]. IVO is an urban land-based robot designed to perform navigation tasks that involve object manipulation for grasping and human–robot interaction. The robot is a modified version of the Tiago robot (https://pal-robotics.com/es/robots/tiago/). In the experiments, for the navigation task, IVO uses a platform with four wheels, a 3D LiDAR where information is filtered to obtain the same observation size as is done in simulations, and a RealSense stereo camera to detect holes and ramps. The implementation for the real experiments was implemented in ROS Melodic.

The real-life experiments were evaluated in three scenarios with static obstacles (see Figure 6). One of the scenarios was used with people to evaluate the robot behavior in case of an unexpected occlusion and how the robot can anticipate a possible collision.

The environment map is used in RVIZ to see the LiDAR lectures, which are used to track people and detect the static obstacles (see Figure 7).

The OP-DDPG model and the hybrid model were tested in the three scenarios.

With one centered obstacle**:** In this simple case, the two models (OP-DDPG and hybrid model) avoid the obstacle, although the hybrid model generates a trajectory far off the obstacle with lower velocities, when the robot is close to the obstacle.With three aligned obstacles**:** This case is probably the most complex one for the robot, due to the confusing space between the obstacles. The robot with the OP-DDPG model attempts to navigate between the static obstacles or navigate in the correct direction, but finally it touches one of the static obstacles. Using the hybrid model, the robot avoids to enter in the reduced space between the static obstacles and navigates toward the left or right space to avoid the obstacles and achieve the goal without collisions.With three obstacles inV-configuration**:** In this case, the robot with the OP-DDPG model sometimes touches one of the static obstacles. The hybrid model achieves the goal without collisions, but it has to reduce its velocity in the space between the three obstacles.With one centered obstacle and a person**:** In this case, a person walks in the opposite direction of the robot, starting from the goal, toward the side initially chosen by the robot to avoid the centered obstacle. When the robot uses the OP-DDPG model, it suddenly detects the person and slightly reduces its velocity. The person has to change its direction to let the robot pass. Using the hybrid model, when the robot detects the person several meters before, the robot changes its direction choosing the other side to avoid collisions and achieves the goal as can be seen in Figure 8.

The quantitative results obtained in the three environments for the success rate and the navigation time (NT) are shown in Table 3. The results are the average values obtained during 10 navigation episodes where the goal distance is 5 m. The hybrid model increases the success rate, but the NT increases as well because the robot reduces its velocity and sometimes oscillates when an obstacle is very close.

### 7.4. Implementation Details

In this subsection, the parameters and features used in our implementation are detailed. All the hyperparameters used in the OP-DDPG model and the hybrid model were taken from [46]:Network hyperparameters: Critic joint, critic observation and actor layer widths are (607,242)×(84)×(241,12,20) with ReLU as the activation function.Reward hyperparameters: As they are ordered in Equation (Equation 3), the reward hyperparameters, $\theta_{r_{P2P}}^{T}$, are (−0.43,0.38,−57.90,0.415,0.67,62.0).

The number of time-step observations taken for training and evaluation at each step is $\theta_{n}=1$. Noise parameters are the described ones in [46]. All results were obtained with the same trained model during two million steps and 500 steps as the maximum episode length.

The implementation was performed in PyTorch using the Gym library for RL. For the simulation environment, the Shapely library was used to manage the floor maps, the robot, and the agent models. The Huber loss was used as the critic loss and Adam as the optimizer. The batch size is 512, and the replay buffer has a capacity of 0.5 million. The remaining training parameters are described in [46]. The moving agents were simulated as circles with a radius of 0.3 m. Agents move to a goal using the SFM with collision prediction [19] to avoid static and moving obstacles in the environment. When an agent goal is reached, a new goal is sampled.

For the real-life experiments, the implementation was done in ROS Melodic, and we used two nodes for the OP-DDPG model: A node is used to filter the frontal LiDAR observations of IVO and the second node is used to calculate the OP-DDPG actions. For the hybrid model, three additional nodes were used: the first node uses the tracker explained in [49] to obtain the pedestrians’ velocities, the second node calculates the SFM forces, and the last node combines the OP-DDPG action with the SFM action. All the nodes were implemented in C++, except the node that computes the OP-DDPG action, that was implemented in Python, using PyTorch and the Gym library. To obtain better results in the real-life experiments, a new model was trained with a radius of 0.45 m for the robot.

## 8. Experiment: UAV Human Accompaniment

In this section, we describe the experimentation process to demonstrate the proper functionality of a UAV accompanying a pedestrian.

### 8.1. Database Generation

Data used for this work was gathered in a simulated environment. The simulation was developed focusing on intuitive controls and visualization, both characteristics tested through a qualitative questionnaire answered by the participants in a first test run.

Therefore, in order to teach our model to behave similarly to a human, we require a significant number of expert demonstrations that will be later used to build a dataset by extracting the relevant features. This dataset will be then used to train the nonlinear model, and hopefully the trained model will fulfill our requirements. Thus, the first step to learn the aerial control policy is to obtain trajectories through human expert demonstrations. The software used to build and run the simulator is FreeGLUT (http://freeglut.sourceforge.net).

From the created expert trajectories we extracted the following features: (i) the companion attractive force, (ii) the goal attractive force, (iii) the pedestrian repulsive force, (iv) the drone velocity, and (v) the *y* component of the static object repulsive force. The two first attractive forces are required to achieve a side-by-side formation with the companion. In order to respect the pedestrian’s personal space the pedestrian repulsive force is necessary. The system also needs the drone velocity to correct motion control. To avoid obstacles, *y* feature is used to keep a proper distance to the floor.

As a result, we obtained a dataset composed by matrices X and Y. X matrix consist of the feature vectors of samples from the expert recorded trajectories. Y matrix are the target vectors of the samples from the expert trajectories. In Figure 9, some environments to train our model are plotted.
(25)$$y_{i}=\left[\begin{array}{ccc} a_{R_{x}} & a_{R_{y}} & a_{R_{z}} \end{array}\right]$$

### 8.2. Simulation Results

The learned flying policies were tested on different test environments. To compute the performance of a model in a test environment, the test environment is simulated, and the trained model is used to compute the acceleration command that must be applied to the drone. The performance metric is computed in each cycle of a test simulation, and thus, the final performance of a certain model in a certain environment is obtained by averaging the instantaneous measurements of performance along a test simulation run on that environment.

In order to evaluate the model through an episode, we use a quantitative metric of performance presented in [50], which is based on the idea of proxemics [6]. We measure the distances of the controlled agent to the human companion and to all other pedestrians and static obstacles at each frame. Then we draw three circles of increasing radius (0.5, 1.5 and 2 m) around the center of the human companion. The smallest represents the area occupied by the robot, the second would be the region we are interested to be in and the third concentric area represents the limits of the zone we want the automated companion to be, being these the limits defined in [6] as the intimate/personal, social and public distances.

We reproduce this evaluation five times for every test environment, Figure 10. We measure the area below the curve for each of them, calculate the average proxemics score per frame through the episode and extract the mean proxemics score per frame and their standard deviation throughout the 10 episodes performed Table 4.

First, as we can observe in the results, the non-linear model score in the simulated environments is clearly higher in comparison to the Linear model. While the Linear model behaves in a reactive way and ends up following the human companion, the Non-Linear is able to learn from human behavior data, recognize such situations, accelerate and reach the human’s side while in motion.

Secondly, it is shown in Table 4 how our method performs better in average than the Linear Model, especially when pedestrians are included in the environment. Throughout every training episode, among the present forces, we were able to register the companion’s repulsive and attractive forces at every frame. This means a great amount of data regarding how to behave around other pedestrians and react to their repulsive forces, which allows the model to better behave in front of other agents.

Finally, these results issuing from successfully navigating through different environments, demonstrate that our method is capable of following a companion operated by an expert, while navigating in environments with pedestrians and static obstacles. These results also demonstrate that our model is able to perform this activity at a human level when it comes to proxemics rules and clearly outperforms the previous method.

### 8.3. Real-Life Experiments

During the real experiments, the learned policy was tested on a real-life environment, which requires the use of a real quadrotor. The selected drone to fill this role is the AR.Drone 2.0 owned by the Institut de Robòtica i Informàtica Industrial [53]. We also made use of the Optitrack Motion Capture system, created by NaturalPoint Inc, which provides all the necessary information about absolute positions of all the elements in the environment that are properly marked. Everything was combined with ROS, and the controller of the drone was implemented as ROS node. With the information provided by Optitrack, the controller node can build the inputs of the model to obtain the predicted acceleration commands for the drone. The drawback of using Optitrack to accurately locate all the elements in the environment is that the working area is limited to 5 × 5 m, which imposes severe constraints on the movement of the drone and the main human, see Figure 11.

First, we built a Optitrack node to detect and localize each element of the environment: the volunteer protective gear, the autonomous Drone and the obstacle of the scene; see Figure 11. With this information, the Optitrack can accurately compute the position of each required element. Figure 12 provides images of the real-life experiments conducted in our laboratory.

When we run several experiments in a simple scenario with the accompanied person and the drone, the behavior of the drone observed in the real-life experiments was correct. During the experiments, we observed that the drone was able to approach the human, and once it reaches a certain distance to the human, it stops and maintains the position in the air. If the human starts moving, the drone is able to follow the person while keeping a safe distance.

In the second set of experiments, a pillar obstacle was introduced into the scenario. We observed that the drone was still able to follow the main human maintaining a safe distance, and additionally it successfully avoided the pillar if it was found in the drone’s way. It is important to mention, however, that, due to the limitations of the experiment setup, it was not possible to fully replicate the simulated results in reality, and we had to run simplified experiments. Figure 12 shows the real-life experiments where the drone moved and was able to accompany a person.

Finally, in Figure 13, we present the proxemics performance results of the real-life experiments. As in the previous section, we measure the area below the curve for each experiment, and we calculate the average proxemics score per frame through different episodes performed. If there are obstacles on the environment the performance decreases; here, the robot cannot achieve high values as the object creates repulsive forces that makes difficult to approach the accompanied person. In contrast, we obtain values over 0.5 if the there are no obstacles.

## 9. Conclusions

The overall goal of the present work is to demonstrate that the combination of learning techniques and Social Force Model, achieve good human-aware (social) navigation trajectories in complex scenarios, where there are environmental constraints (buildings, static objects, etc.), pedestrians and other mobile vehicles. This work presents two new techniques that combine machine learning and SFM: (i) social robot navigation using deep RL and SFM; and (ii) accompaniment of a pedestrian using supervised learning and 3D SFM. The first method was tested with a ground robot and the second one with an aerial robot.

Sometimes, in deep RL, it is difficult to obtain a good performance in complex situations that require a great deal of parameter tuning, reward shaping and other methods that do not guarantee an improvement in the results. The use of SFM was demonstrated to be very useful to simulate people’s behavior in many situations. The results obtained in simulations and experiments with the IVO robot demonstrate that the SFM is a useful technique to improve RL for robot navigation.

The ability for the robot to avoid obstacles was increased in all cases when the SFM was added during simulations and generates safer policies in real-life environments with people. The hybrid model provides a robust policy against noise in simulations and real experiments with very little information about the environment. Moreover, regarding the teaching model for an aerial robot, the learning by demonstration architecture obtained in simulated environments by an expert user can be tested in the same type of simulated environments, and the trained architecture provided good results when it was applied on a real drone in real-life experiments.

This research reveals that it is possible to learn flying behaviors by the use of recorded aerial trajectories performed by an expert pilot. Furthermore, we may resolve that the system learns better polices with the use of simple scenarios instead of complex environments. Thus, our drone was capable of learning a control policy and navigating close to a person in the working environment. Finally, the authors believe that the combination of SFM with different machine learning techniques enhance the performance of the presented frameworks, achieving a human-aware (social) navigation of both ground robots and aerial robots.

## Figures and Tables

**Figure 1 sensors-21-07087-f001:**
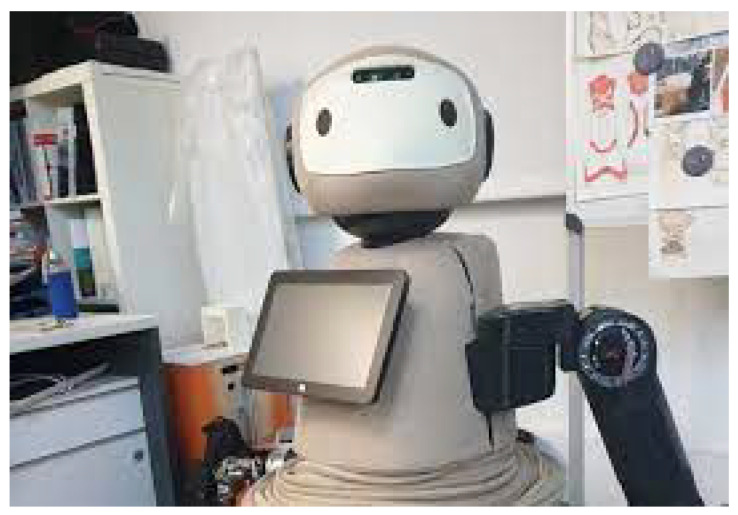
IVO Robot. New design of a humanoid robot for citizen assistance.

**Figure 2 sensors-21-07087-f002:**
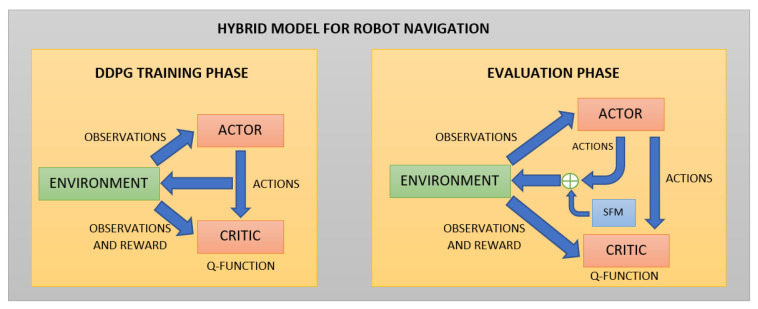
Hybrid Model Scheme. The model is trained using the DDPG algorithm without SFM actions. Once the model is trained in the evaluation phase, the SFM actions are added to move the robot in the environment.

**Figure 3 sensors-21-07087-f003:**
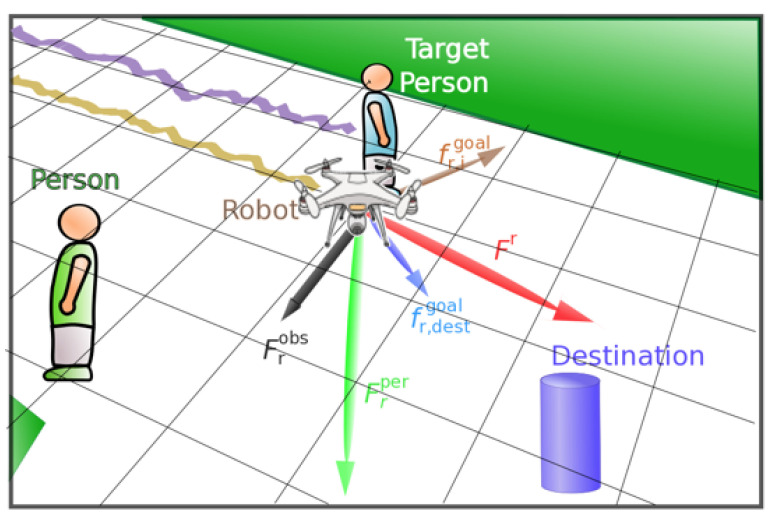
SFM, forces applied to the drone while accompanying a person.

**Figure 4 sensors-21-07087-f004:**
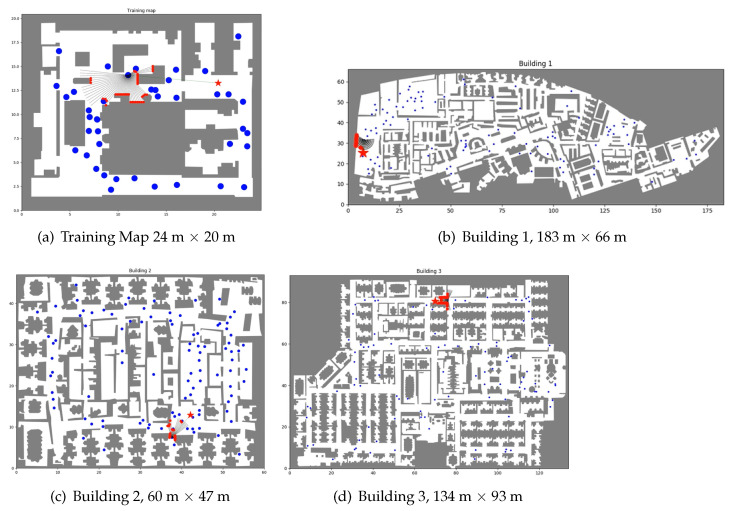
Floor maps. These are the floor maps for training and evaluation with their sizes in meters.

**Figure 5 sensors-21-07087-f005:**
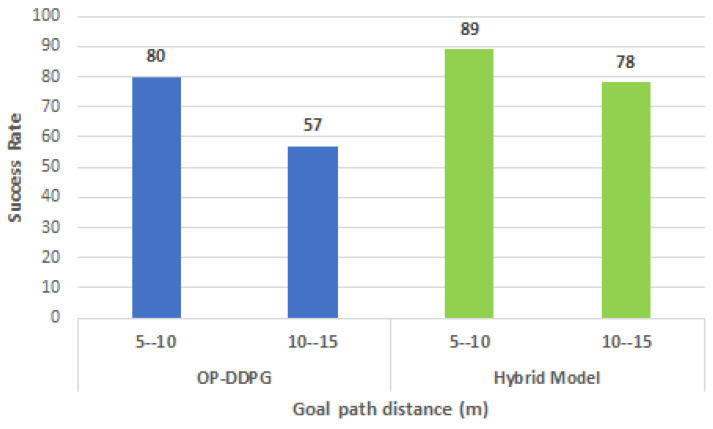
Success rate variation with distance in Building 2. The graphic shows less rate decay with path distance in the hybrid model (green bars) with 20 moving agents.

**Figure 6 sensors-21-07087-f006:**
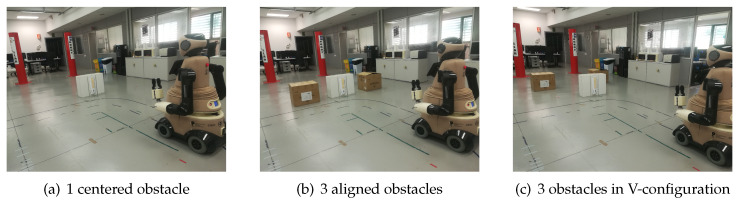
Real-life environments for experiments. In all the cases the robot goal is between the red columns.

**Figure 7 sensors-21-07087-f007:**
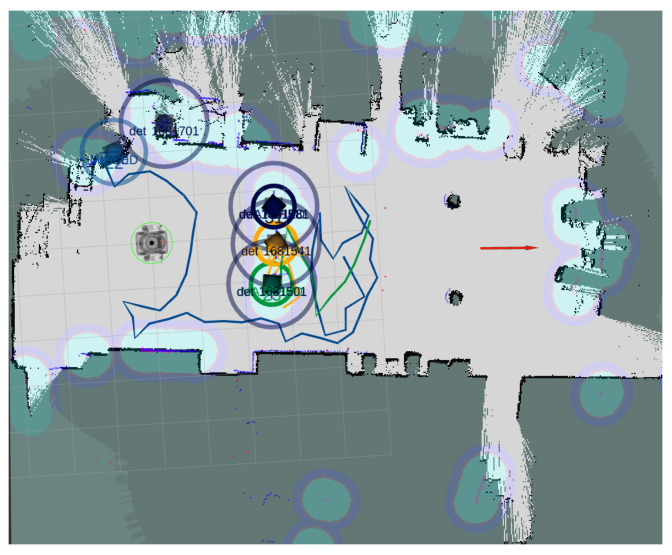
RVIZ representation. In this real time representation we can see the LiDAR observations (small red points), the tracks for moving obstacles or people (big circles) and the goal position (red arrow).

**Figure 8 sensors-21-07087-f008:**
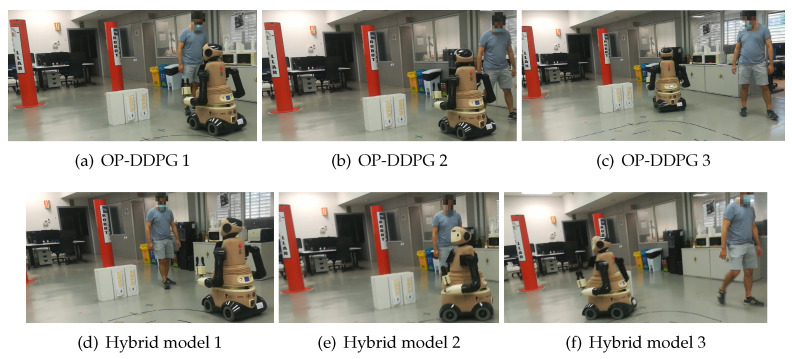
Experiment with a person. From 1 to 3 shows the robot behavior when the robot uses the OP-DDPG model and the hybrid model. In the first case, the person has to adapt to the robot motion, while, in the second case, it is the robot who adapts to the person.

**Figure 9 sensors-21-07087-f009:**
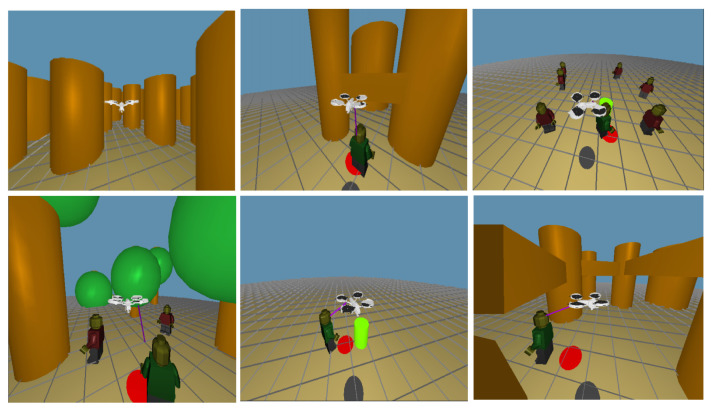
Simulated environments: Frames from the simulator’s view for different environments.

**Figure 10 sensors-21-07087-f010:**
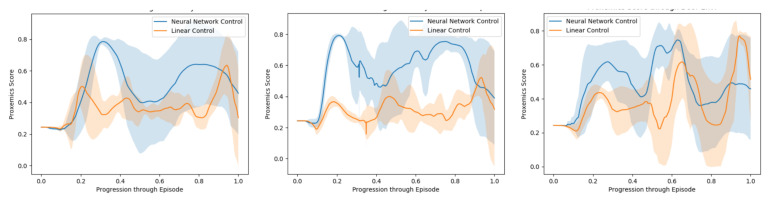
Evaluation Simulation. Evaluation of the model through proxemics social distances as presented in Section 6. From left to right: Cluttered environment, crowded scenarios and cluttered and crowded scenarios.

**Figure 11 sensors-21-07087-f011:**
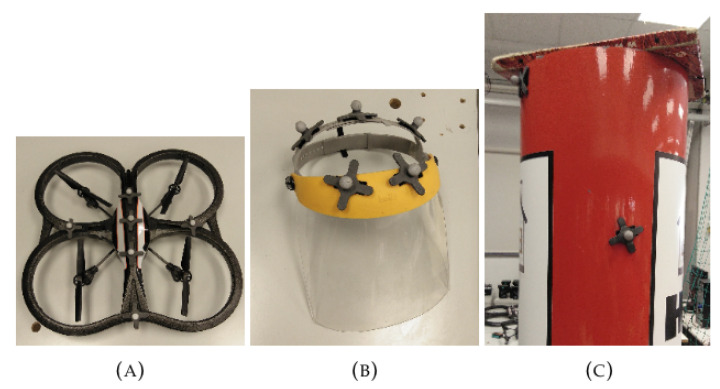
Hardware involved: SMarkers applied to (**A**) the AR.Drone, (**B**) the main human protective gear and (**C**) the obstacle.

**Figure 12 sensors-21-07087-f012:**
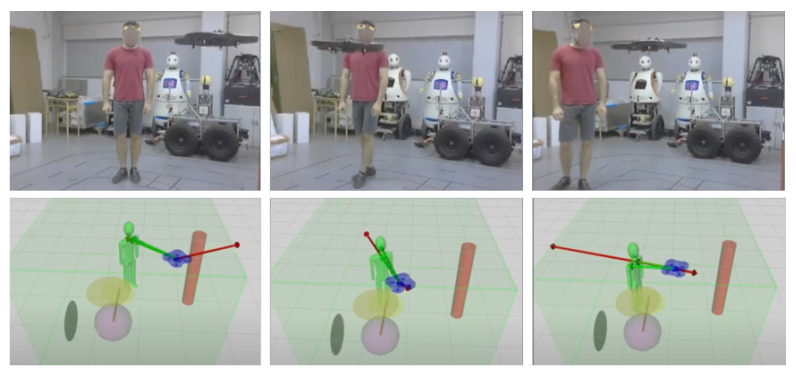
Real-life experiments: (**Top**): Validation of the model in a real-world environment. (**Bottom**): Gazebo simulation replicating the conditions found in the laboratory. The drone, the main human and an obstacle are the only elements in the environment, which is also limited by the green volume.

**Figure 13 sensors-21-07087-f013:**
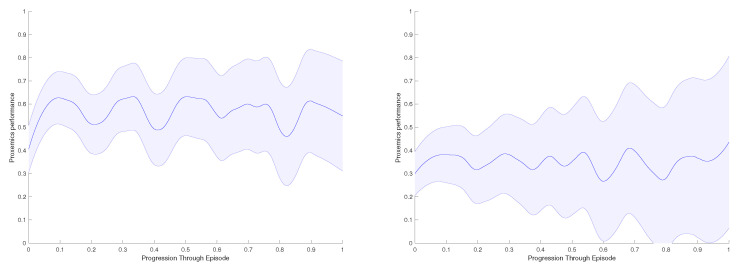
Real-life experiment with a person: Evaluation Results of the model through proxemics social distances as presented in Section 6. (**Left**): Environment with no obstacles. (**Right**): Environment with one obstacle.

**Table 1 sensors-21-07087-t001:** OP-DDPG and Hybrid Model comparison in static environments for Euclidean and path distance cases.

Environment	OP-DDPG			Hybrid Model		
	SR	CR	TOUT	SR	CR	TOUT
**Building 1 (ED)**	60	34	6	66	11	23
**Building 2 (ED)**	50	46	4	74	15	11
**Building 3 (ED)**	65	35	0	81	11	8
**Building 1 (PD)**	75	11	14	88	3	9
**Building 2 (PD)**	87	13	0	96	1	3
**Building 3 (PD)**	87	12	1	97	1	2

**Table 2 sensors-21-07087-t002:** OP-DDPG and Hybrid Model comparison for Euclidean and path distance cases in dynamic environments with 20 and 100 moving agents.

Environment-Agents	OP-DDPG			Hybrid Model		
	SR	CR	TOUT	SR	CR	TOUT
**Building 1 ED-20**	58	37	5	67	22	11
**Building 2 ED-20**	47	44	9	66	22	12
**Building 3 ED-20**	65	34	1	77	18	5
**Building 1 PD-20**	74	11	15	84	4	12
**Building 2 PD-20**	80	20	0	89	3	8
**Building 3 PD-20**	84	16	0	88	2	10
**Building 1 ED-100**	52	44	4	61	27	12
**Building 2 ED-100**	39	50	11	53	37	10
**Building 3 ED-100**	57	41	2	71	22	7
**Building 1 PD-100**	67	16	17	84	3	13
**Building 2 PD-100**	61	39	0	76	16	8
**Building 3 PD-100**	83	17	0	92	0	8

**Table 3 sensors-21-07087-t003:** OP-DDPG and Hybrid Model comparison for real-life experiments in the three environments.

Environment	OP-DDPG		Hybrid Model	
	SR	NT(s)	SR	NT(s)
**1 centered obstacle**	90	18.40	100	29.80
**3 aligned obstacle**	20	34.50	80	40.80
**3 obstacles in V-configuration**	50	17.85	100	36.20
**1 centered obstacle and 1 person**	80	28.45	100	29.01

**Table 4 sensors-21-07087-t004:** Nonlinear model performance against the previous approach, linear model. Evaluation done through the proxemics social distance metric. Bold shows the best value.

	Cluttered Env.		Crowded Env.		Cluttered & Crowded Env.	
Models	Mean	Std	Mean	Std	Mean	Std
**Linear**	0.480	0.066	0.515	0.097	0.533	0.084
**Non-Linear**	**0.616**	0.119	**0.586**	0.0123	0.542	0.137

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
