# Peer review of "Social Robot Navigation Tasks: Combining Machine Learning Techniques and Social Force Model"

_sensors, 2021, doi:10.3390/s21217087_

Round 1

Reviewer 1 Report

The paper is of interest and it is devoted to the actual problem but a recommend to significantly correct it. My main remarks concerns the clear goal of the research, structure and results.

These are my remarks.

I. I have some remarks to the abstract:

1. Please supplement the abstract by the main goal of this research. What is the main problem you try to decide?

2. The Abstract contain too many details but not conclusions. 

II. I have a major remark to the structure of the paper. At the present time it seems disordered. The Problem formulation appears only at the subsection 4.1 at the 7 page from 22! Moreover another "Problem definition" appears at the 9th page. For the convenient reading and maintaining the style of scientific paper this text should be significantly restructured and I recommend you to hold the exemplarily scheme:

Introduction with the Problem formulation, Related Works, Your proposed methods and algorithms (with models, equations and so on), Experimtntal setup (it is very important to describe all data of experiments), Test data, Results, Conclusion.

III. Bibliography contain a number of stylistic misprints:

1. Not all years in the bibliography are bolded.

2. Some references have "pages" (for example, see "pp." in the 29 reference), but some is not (for example, "529-533" in the 31 reference). Please unify.

3. Please read the bibliography once again and correct possible misprints.

IV. Another remarks:

1. Please supplement the "Institut de Robotica i Informatica Industrial" with the reference. Without any reference this expression in the scientific paper is not factful.

2. I guess the first proposition in the Conclusions is inadmissible. The goal of the paper is not to try some methods or their combination. The goal of scientific paper is to SOLVE THE PROBLEM or to investigate it for its subsequent solution.

3. The first mention of "Social Force Model" do not have abbreviation but the  next inclusions have it and repeat many times (Even in Conclusions). Please keep one abbreviation. Moreover the reference for SFM appears only at the 4th page. It is inapproproate. It should be writted at the first mention of this term.  

4. The similar case with the AutoRL. The first mention on the 4th page is not have an abbreviation, but it appears on the 6th page.

5. The subsection 7.3 contain only verbal results. It will be reliability to supplement by the numerical data.

Author Response

Dear reviewer,

We thank the reviewers for their comments and advice. Also, we appreciate the opportunity to resubmit our manuscript. It has helped to improve the quality of the manuscript. In this document, we would like to answer their main remarks and suggestions. Changes in the submitted paper appear in red color.

Comments and Suggestions for Authors

The paper is of interest and it is devoted to the actual problem but a recommend to significantly correct it. My main remarks concerns the clear goal of the research, structure and results.

These are my remarks.

I have some remarks to the abstract:

  1. Please supplement the abstract by the main goal of this research. What is the main problem you try to decide?

Answer: Main goals has been included in the abstract as well as the definition of the problem to be solved.

  1. The Abstract contains too many details but not conclusions.

Answer: Part of the abstract has been rewritten following the suggestions of the reviewers.

  1. I have a major remark to the structure of the paper. At the present time it seems disordered. The Problem formulation appears only at the subsection 4.1 at the 7 page from 22! Moreover another "Problem definition" appears at the 9th page. For the convenient reading and maintaining the style of scientific paper this text should be significantly restructured and I recommend you to hold the exemplarily scheme:

Introduction with the Problem formulation, Related Works, Your proposed methods and algorithms (with models, equations and so on), Experimtntal setup (it is very important to describe all data of experiments), Test data, Results, Conclusion.

Answer: We thank the reviewer for the suggestions on the structure of the paper. We followed his/her advice, and we created a new section (Section 3) where we introduce the problem definition.

III. Bibliography contain a number of stylistic misprints:

  1. Not all years in the bibliography are bolded.

Answer: All the years have been bolded

  1. Some references have "pages" (for example, see "pp." in the 29 reference), but some is not (for example, "529-533" in the 31 reference). Please unify.

Answer: All the references now have “pp.”

  1. Please read the bibliography once again and correct possible misprints.

Answer: We revised all the references and corrected them.

Another remarks:

  1. Please supplement the "Institut de Robotica i Informatica Industrial" with the reference. Without any reference this expression in the scientific paper is not factful.

Answer: The Institut de Robotica i Informatica Industrial website has been added as reference.

  1. I guess the first proposition in the Conclusions is inadmissible. The goal of the paper is not to try some methods or their combination. The goal of a scientific paper is to SOLVE THE PROBLEM or to investigate it for its subsequent solution.

Answer: It can be found on the text that we rewrite part of the conclusion sections following the suggestions of the reviewer.

  1. The first mention of "Social Force Model" do not have abbreviation but the next inclusions have it and repeat many times (Even in Conclusions). Please keep one abbreviation. Moreover the reference for SFM appears only at the 4th page. It is inapproproate. It should be writted at the first mention of this term.

Answer: The abbreviation SFM has been added in the first inclusion of the term. The same is applied to the abbreviations RL and ESFM.

  1. The similar case with the AutoRL. The first mention on the 4th page is not have an abbreviation, but it appears on the 6th page.

Answer: The complete term for the abbreviation AutoRL has been added on the 4th page.

  1. The subsection 7.3 contain only verbal results. It will be reliability to supplement by the numerical data.

Answer: Two numerical graphs and one table have been included in the text to demonstrate the reliability of the real-life experiments.

Reviewer 2 Report

  • spelling errors are found in the text, a thorough revision is recommended.
  • some authors consider reinforcement learning as an intermediate point between supervised and unsupervised learning, However, the authors refer first as unsupervised and then as autoRL, it would be recommended that they homogenize the terms
  • In line 240 the authors refer to the reward function (3), however, this function seems to be wrong, the reward function is defined as a scalar function and equation (3) expresses vector magnitudes
  • It is recommended that the authors explain how they obtained the proposed reward function, since it depends on avoiding suboptimal action policies.
  • it is recommended to make clear the main contribution of the article in the conclusions

Author Response

Dear reviewer,

We thank the reviewers for their comments and advice. Also, we appreciate the opportunity to resubmit our manuscript. It has helped to improve the quality of the manuscript. In this document, we would like to answer their main remarks and suggestions. Changes in the submitted paper appear in red color.

Comments and Suggestions for Authors

â—Ź    spelling errors are found in the text, a thorough revision is recommended.

Answer: We reviewed all the paper and correct the spelling errors.

â—Ź    some authors consider reinforcement learning as an intermediate point between supervised and unsupervised learning, However, the authors refer first as unsupervised and then as autoRL, it would be recommended that they homogenize the terms

Answer: The unsupervised learning references have been eliminated and AutoRL is described as a reinforcement learning method to homogenize and clarify the terms.

â—Ź    In line 240 the authors refer to the reward function (3), however, this function seems to be wrong, the reward function is defined as a scalar function and equation (3) expresses vector magnitudes

Answer: The reference (3) was wrong and it was changed by the reference (13).

â—Ź    It is recommended that the authors explain how they obtained the proposed reward function, since it depends on avoiding suboptimal action policies.

Answer: In line 255, it is explained that the reward function is the one obtained in [39] work. In lines 506 and 507, it is explained that the reward hyperparameters have been taken from [47].

â—Ź    it is recommended to make clear the main contribution of the article in the conclusions

Answer: It can be found on the text that we rewrite part of the conclusion sections following the suggestions of the reviewer.

Round 2

Reviewer 1 Report

All my comments have been taken into account.